# Macrophages as a Source and Target of GDF-15

**DOI:** 10.3390/ijms25137313

**Published:** 2024-07-03

**Authors:** Lina Susana Silva-Bermudez, Harald Klüter, Julia G. Kzhyshkowska

**Affiliations:** 1Institute of Transfusion Medicine and Immunology, Medical Faculty Mannheim, Heidelberg University, 68167 Mannheim, Germany; linasusana.silvabermudez@medma.uni-heidelberg.de (L.S.S.-B.); harald.klueter@medma.uni-heidelberg.de (H.K.); 2German Red Cross Blood Service Baden-Württemberg-Hessen, 68167 Mannheim, Germany

**Keywords:** cytokine, growth factor, receptor, inflammation, healing, fibrosis

## Abstract

Growth differentiation factor 15 (GDF-15) is a multifunctional cytokine that belongs to the transforming growth factor-beta (TGF-β) superfamily. GDF-15 is involved in immune tolerance and is elevated in several acute and chronic stress conditions, often correlating with disease severity and patient prognosis in cancer172 and metabolic and cardiovascular disorders. Despite these clinical associations, the molecular mechanisms orchestrating its effects remain to be elucidated. The effects of GDF-15 are pleiotropic but cell-specific and dependent on the microenvironment. While GDF-15 expression can be stimulated by inflammatory mediators, its predominant effects were reported as anti-inflammatory and pro-fibrotic. The role of GDF-15 in the macrophage system has been increasingly investigated in recent years. Macrophages produce high levels of GDF-15 during oxidative and lysosomal stress, which can lead to fibrogenesis and angiogenesis at the tissue level. At the same time, macrophages can respond to GDF-15 by switching their phenotype to a tolerogenic one. Several GDF-15-based therapies are under development, including GDF-15 analogs/mimetics and GDF-15-targeting monoclonal antibodies. In this review, we summarize the major physiological and pathological contexts in which GDF-15 interacts with macrophages. We also discuss the major challenges and future perspectives in the therapeutic translation of GDF-15.

## 1. Introduction

Macrophages are ubiquitous in almost all human tissues and profoundly influence the healing and remodeling of damaged tissue. At the core of their impressive versatility lies their ability to rapidly polarize in response to stimuli. For example, during healing processes, macrophages remove local debris via phagocytosis and secrete various factors that promote angiogenesis and scar formation through auto- and paracrine mechanisms [1,2]. This dynamic adaptability to their environment has made macrophages essential to the body’s natural tissue maintenance mechanisms [3].

During the inflammatory phase of injury, there is an increase in macrophage infiltration, accompanied by an enhanced production of proinflammatory cytokines. This, in turn, induces the recruitment and proliferation of macrophage progenitor cells. A phenotypic transition to M2 favors the resolution of inflammation through the secretion of IL-10, transforming growth factor β (TGF-β), and vascular endothelial growth factor (VEGF), which supports fibroblast proliferation and promotes angiogenesis. The macrophage infiltration then decreases until wound resolution progresses [4,5].

The ontogeny of macrophages may influence their function during healing processes. Tissue-resident macrophages can originate from resident cells that locally proliferate, from a spleen monocyte reservoir, or from blood peripheral circulating bone marrow-derived monocytes (BMDM) [6]. The initial proinflammatory response seems to be supported by spleen-derived monocytes that differentiate into M1. In contrast, the resolution-like M2 phenotype appears to be derived from resident macrophages and circulating monocytes [6].

On the opposite end of the healing spectrum lies fibrosis, a process resulting from excessive extracellular matrix (ECM) accumulation and defective remodeling [7]. Macrophages contribute to fibrogenesis by recruiting and activating fibroblasts, secreting TGF-β1, and modulating the tissue microenvironment. They also contribute to the resolution of fibrosis by secreting matrix metalloproteinases (MMPs) that degrade the ECM, such as MMP2 and MMP13, and by participating in the clearance of senescent cells [8]. The balance between reparative processes and fibrotic responses is regulated by the interplay of local inflammatory cues. For instance, members of the TGF-β superfamily exert dualistic effects, depending on their content, concentration, spatial and temporal production, and clearance, as well as the activation status of target cells [9].

Several years ago, reprogramming of macrophages aroused a new wave of interest, driven primarily by the need to enhance the efficacy of conventional chemotherapy and novel immunotherapy in solid tumors [10,11,12]. The principles of reprogramming, initially developed for tumor-associated macrophages (TAMs), can be further utilized for therapeutic control of macrophage activity in patients with cardiometabolic diseases and in regenerative medicine. Successful reprogramming requires an understanding of the key cytokines that are able to regulate the switch from an inflammatory to a healing phenotype in macrophages and in the tissue microenvironment. The accumulated experimental data indicate that GDF-15 is such a cytokine.

Our review focuses on growth differentiation factor 15 (GDF-15), a member of the transforming growth factor-beta (TGF-β) superfamily, which can be both produced and cleared by macrophages. GDF-15 production can be induced by inflammatory mediators, yet the majority of its effects are anti-inflammatory and pro-fibrotic. We have summarized the available data on the role of GDF-15 in pathologies in which monocytes and macrophages play a major role.

## 2. GDF-15 Is a Member of the TGF-β Superfamily

GDF-15, also known as MIC-1, PTGF-β, PDF, and NAG-1, is a multifunctional cytokine and is a distant member of the glial cell-derived neurotrophic factor (GDNF) family and the TGF-β superfamily [13,14,15]. Numerous TGF-β family members are known to be produced by macrophages and to target macrophages, including TGF-β1, GDFN, and BMP-2 [16,17,18]. GDF-15 was discovered in the late 1990s, when it was identified as an autocrine cytokine capable of reducing proinflammatory macrophage activation after lipopolysaccharide (LPS) stimulation in the human myelomonocytic cell line U93737. In parallel, Lawton et al. discovered its expression in the placenta during early and late gestation [19]. Moreover, Paralkar et al. found elevated expression of this protein in the prostate and named it prostate-derived factor (PDF) [20]. A few years later, Baek et al. observed the upregulation of GDF-15 in HCT-116 colon cancer cells upon treatment with non-steroidal anti-inflammatory drugs (NSAIDs) [21].

The GDF15 gene is found throughout the animal kingdom and is well-conserved in vertebrates, particularly in mammals [22,23]. It is located on chromosome 19p12-13.1 and consists of two exons (309 bp and 891 bp) separated by a 2.9 kb intron [19]. Analogous to other members of the TGF-β family, GDF-15 has a dimeric disulfide-bonded configuration and is synthesized as a proprotein [13]. Its general structure includes a propeptide followed by an RXXR furine-like site and the mature protein. GDF-15 conserves the seven cysteine domains typical of the TGF-β family, sharing a 20% amino acid identity that gives rise to its cysteine knot crystal motif (Figure 1a,c) [15]. In addition to its proximity to the TGF-β family, GDF-15 resembles the structure of the GDNF family ligands, sharing 16% amino acid identity (Figure 1b,c) [13]. GDF-15’s immature form consists of 308 amino acids, including a 9-amino-acid signal peptide, a 167-amino-acid propeptide, and a 112-amino-acid mature protein. As part of its post-translational modifications, the immature form undergoes proteolytic cleavage, leaving the mature (13 kD) and propeptide (30 kD) forms, which are later cleaved at the RXXR furine-like site [15]. This site is recognized and cleaved by proprotein convertase subtilisin/kexin type (PCSK)-3, -5, and -6 or by MMPs, facilitating GDF-15 maturation [24,25,26]. After dimerization, the mature form, consisting of 224 amino acids (25 kD), and the propeptide are secreted. In contrast to the general structure of the TGF-β family, the propeptide of GDF-15 lacks cysteine residues [15]. Interestingly, the propeptide appears to function autonomously [27]. Latent stromal stores of this immature form have been found in various tissues and pathologies, such as in prostate cancer. These stores serve as a reservoir for GDF-15 [28]. Moreover, the secretion of the propeptide is faster than that of the mature GDF-15 [29].

### 2.1. Producing Cells and Regulators of GDF-15 Expression

Under physiological conditions, GDF-15 is produced at high levels by trophoblasts in the terminal villi of the placenta, with levels reaching up to 54,000 pg/mL [31,32]. Other human tissues produce lower levels of GDF-15, including stomach, skeletal muscle, subcutaneous adipose tissue, prostate epithelium, bladder, kidney, and lung [33,34,35,36,37,38,39,40]. Furthermore, GDF-15 has been found in most human fluids, including blood, amniotic fluid, bronchoalveolar fluid, and cerebrospinal fluid [31,37] (Figure 2).

A diverse array of cell types, such as fibroblasts, adipocytes, macrophages, and epithelial and endothelial cells contribute to the production of GDF-15 (Table 1). However, most of them express GDF-15 under stress conditions, such as exposure to LPS, bleomycin, and oligomycin [41,42,43]. GDF-15 production is specialized in some organs, such as in the stomach, where it is produced by chief cells, located at the bottom of the gastric glands [40]. Among the blood cells, the myeloid population exhibits higher expression levels of GDF-15 compared to the lymphoid population. Following clodronate-induced depletion of myeloid cells in mice, GDF-15 mRNA levels significantly dropped, highlighting a major myeloid contribution to systemic GDF-15 levels [44]. During the hematopoietic differentiation of CD34+ progenitors, erythroid precursors secrete high levels of GDF-15. In contrast, the myeloid cell lineage exhibited minimal levels of GDF-15 during maturation [45]. During erythroblast maturation, GDF-15 levels are found up to 74,000 pg/mL [46]. Recently, megakaryocytes were shown to be key cells expressing GDF-15 in the bone marrow [47].

**Table 1 ijms-25-07313-t001:** GDF-15-producing cells.

Primary Cells	Species	Conditions	Measurement	Reference
Adipose tissue macrophages	Human	Obesity	RT-PCR	[44]
		Rosiglitazone-treated	Microarray	[48]
Human nasal epithelial cell	Human	LPS treatment	RT-PCR	[49]
Lung fibroblasts	Mouse	Bleomycin treatment	RT-PCR	[50]
Cardiomyocytes	Rat	Stretch conditions	RT-PCR	[51]
Chief cells	Rat/Human	Obesity	Immunohistochemistry	[40]
Erythroid cells	Human	Maturation	RT-PCR	[52]
Megakaryocytes	Human	Primary myelofibrosis	Immunohistochemistry	[47]
Primary erythroblasts	Human	7 and 14 d	RT-PCR, ELISA	[46]
Hepatocytes	Human	Metformin treatment	RT-PCR	[53]
Cell lines				
Macrophages-like cells				
THP-1	Human	LPS treatment	RT-PCR	[54]
RAW264.7	Mouse			
Endothelial cells				
Endothelial colony-forming cells generated from blood	Human	7 d	RT-PCR, ELISA	[55]
Human aortic endothelial cells	Human	hrCRP	RT-PCR, ELISA	[56]
Connective and soft tissue cells				
Adipocytes	Human	SAT-differentiated	RT-PCR, ELISA	[41]
	Human	Oligomycin treatment	ELISA	
	Mouse	rIL-4 and rIL-13	RT-PCR, ELISA	[57]
Embryonic adipocyte-like cell line (3T3-L1)	Mouse	Tunicamycin treatment	RT-PCR	[42]
Myoblasts (C2C12)	Mouse	.	RT-PCR, Western blot	[58]
Trophoblastic cells (BeWo)	Human	24 h	ELISA	[31]
Cancer cells				
Alveolar basal epithelial cells (A549)	Human	Bleomycin treatment, C5a treatment	RT-PCR	[43,59]
Colorectal cancer cells (HCT-116)	Human	NSAID, Indomethacin	Northern/Western blot	[21]
Hypopharyngeal carcinoma cells (FaDu cells)	Human	Head and neck cancer	Western blot	[60]
Prostate cancer cells (LNCaP-C81 cell line)	Human	Prostate cancer	Western blot	[61]

CRP: C-reactive protein; hr: human recombinant; GDF-15: growth differentiation factor 15; ELISA: enzyme-linked immunosorbent assay; LPS: lipopolysaccharide; NSAID: non-steroidal anti-inflammatory drug; RT-PCR: reverse transcription real-time polymerase chain reaction; SAT: Subcutaneous adipose tissue; THP-1: human acute monocytic leukemia cell line.

Several transcriptional factors have been identified that induce GDF-15 expression (Table 2). For example, upon exposure to C-reactive protein (CRP), p53 binds to the GDF-15 promoter and induces GDF-15 transcription in human aortic endothelial cells [56]. This highlights an association between the two biomarkers, GDF-15 and CRP, which are co-elevated in acute inflammatory conditions and also induce GDF-15 expression following damage induced in enterocytes. In this context, GDF-15 functions as a pro-apoptotic factor and triggers the expression of activating transcription factor 3 (ATF3), a pro-survival protein induced during cellular stress [59]. The interaction between ATF3 and GDF-15 also been reported in human colorectal cancer cells [62].

Transcription factor EB (TFEB), a regulator of energy expenditure and an inducer of autophagy, binds to the GDF-15 promoter and increases GDF-15 expression following exposure to lysosomal stressors in adipose tissue macrophages. This mechanism mediates a reduction in metabolic inflammation during high-fat-induced obesity in mice [44]. In IL-13-treated adipocytes, signal transducer and activator of transcription 6 (STAT6) induces the expression of GDF-15. This process seems to improve glucose tolerance [57]. In addition, under mitochondrial stress, C/EBP homologous protein promotes GDF-15 expression in murine myoblasts and hepatocytes [58]. Another known transcription factor is early growth response 1 (EGR1), which forms a positive feedback loop with GDF-15 and acts as a promoter for EGR1 [60]. PPARγ and RXRα are regulatory elements of the GDF15 locus in monocyte-derived macrophages that are active during muscle regeneration [63]. Nuclear factor erythroid 2-related factor 2 (NRF2), a factor involved in the transcription of antioxidant proteins, also promotes GDF-15 expression in BMDMs treated with NSAIDs [64]. Table 2 highlights the diverse range of GDF-15 inducers.

**Table 2 ijms-25-07313-t002:** Transcription factors regulating GDF-15 expression.

Transcription Factors	Expressed in Macrophages	Cells	Conditions	Reference	
	ATF4	+	Human nasal epithelial cells	LPS treatment	[49]
			Murine hepatocytes	Metformin treatment	[53]
			Murine embryonic adipocyte-like cell line (3T3-L1)	Tunicamycin treatment	[42]
	CHOP	+	PMA-differentiated THP-1, PBMCs	SFAs treatment	[65]
			Murine C2C12 myoblasts	CRIF1 deficiency	[58]
			Murine hepatocytes	Metformin treatment	[53]
EGR1	+	Hypopharyngeal carcinoma cell line (FaDu), human epithelial carcinoma cell line (KB)	.	[60]	
KLF5	+	Adenocarcinoma alveolar basal epithelial cells (A549)	C5a treatment	[59]	
NF-κB	+	Immortalized mouse embryonic Fibroblasts	.	[66]	
NRF2	+	Murine and human BMDM	NSAIDs	[64]	
p53	+	Human aortic endothelial cells	CRP supplementation	[56]	
Sp1, Sp3	+	Human colorectal cancer cells (HCT-116)	.	[21]	
STAT6	+	Murine adipocytes	IL-13 treatment	[57]	
TFEB	+	Human and murine adipose tissue macrophages	Obesity	[44]	
YAP *	−	Breast cancer cells and cytotrophoblast	.	[67]	

* YAP has been shown to act as a negative regulator of GDF-15 expression. ATF: activating transcription factor; BMDM: blood peripheral circulating bone marrow-derived monocytes; CHOP: C/EBP homologous protein; CRIF1: Cytokine response 6 (CR6)-interacting factor 1; CRP: C-reactive protein; EGR1: early growth response 1; KLF: Kruppel-like factor; NF-κB: nuclear factor kappa-light-chain-enhancer of activated B cells; NRF2: Nuclear factor erythroid 2-related factor 2; NSAID: non-steroidal anti-inflammatory drug; PBMC: peripheral blood mononuclear cells; PMA: phorbol 12-myristate 13-acetate; SFA: saturated fatty acids; STAT6: signal transducer and activator of transcription 6; TFEB: transcription factor EB; THP-1: human acute monocytic leukemia cell line; YAP: Yes-associated protein.

### 2.2. Known Receptors and Target Cells of GDF-15

In 2017, GDNF family receptor α-like (GFRAL), a member of the GDNF receptor α family, was identified as a receptor for GDF-15. This receptor is highly expressed in the area postrema and nucleus of the solitary tract of the hindbrain in mice, non-human primates, and humans [68,69,70]. Using an unbiased ligand–receptor coupling approach, Mullican et al. and Yang et al. found that GDF-15 binds exclusively to GFRAL, excluding high-affinity binding to other receptors, including those of the TGF-β receptor family [69,70]. Other potential ligands for GFRAL, such as TGF-β and GDNF-similar ligands (GFLs), were also ruled out, highlighting an exclusive partnership between GDF-15 and GFRAL [69]. Together with Emerson et al., these three research groups dissected the mechanism of GDF-15–GFRAL binding and its effect on appetite regulation [70]. Moreover, the receptor tyrosine-protein kinase (RET) was found to be a co-receptor necessary for the metabolic effects of GDF-15 through GFRAL [68,70]. Upon its binding, GDF-15 promotes the physical interaction between GFRAL and RET and mediates the activation of RET phosphorylation and the intracellular phosphorylation cascade of extracellular-signal-regulated kinase, also called EKR, Akt, and phospholipase C [68,69].

Like GFRAL, RET expression is particularly high in the area postrema and nucleus of the solitary tract in the hindbrain of both mice and non-human primates. Beyond this, GFRAL is also expressed in the human spleen, thymus, testis, and adipose tissue, as well as in isolated adipocytes, but not in macrophages [35,43]. RET is expressed in most mouse tissues except liver, kidney, and adrenal glands [69]. GFRAL/RET expression was also shown in osteoblasts in a murine model of prostate cancer bone metastasis [61]. In the immune cells, GFRAL is expressed by regulatory T cells (Treg) treated with recombinant human GDF-15 (rGDF-15) [71]. These findings highlight that GFRAL expression extends beyond the boundaries of the central nervous system, possibly increasing in response to pathological stimuli, and varies among different cell types.

To date, other receptors have been investigated as possible mediators of GDF-15 peripheral actions. For instance, rGDF-15 was shown to increase the phosphorylation of downstream components of the TGF-βI receptor cascade, namely Smad2/3, in fibroblasts and in human colorectal carcinoma cells, which was associated with metastasis induction [43,72]. In THP-1 cells and neutrophils, rGDF-15 binds to the TGF-βI receptor and effectively blocks the cytokine-induced phosphorylation cascade, promoting anti-inflammatory activity [73]. However, concerns have been raised about the validity of these results due to reported TGF-β contamination of rGDF-15 [74]. Other alternative mechanisms could explain the peripheral effects of GDF-15, such as the existence of a soluble GFRAL that would form a complex with GDF-15 and facilitate its recognition and internalization in further tissues [75]. Another consideration is that GDF-15 may bind to other receptors depending on the cell type and underlying pathological condition. Therefore, the involvement of TGF-β receptors or other receptors in the response of cells to GDF-15 is still controversial.

Using His-tag mediated pull-down assays and mass spectrometry, Wang et al. identified CD48 as a receptor for GDF-15 in Jurkat T cells. The interaction between GDF-15 and CD48 leads to inhibition of CD48 cascade pathways, resulting in downregulation of STUB1 and subsequent accumulation of FOXP3, thereby facilitating Treg activation. This binding was found to be exclusive to GDF-15 and not to TGF-β [76].

GDF-15 is also recognized as a common endocytic ligand for both stabilin-1 and -2 [77]. Stabilin-1 (STAB1, FEEL-1, CLEVER-1, KIAA0246) is a multifunctional scavenger receptor that mediates the endocytic and phagocytic internalization of various unwanted self-ligands, thereby contributing to tissue homeostasis [78]. Stabilin-1 expression is observed in sinusoidal endothelial cells within the spleen, liver, and lymphatic vessels. It is also present in resident macrophages, excluding Kupffer cells, and in TAMs. Macrophages in inflamed or healing cardiac tissue also show expression of stabilin-1 [79,80,81,82,83,84]. Macrophages express stabilin-1 in response to IL-4 and dexamethasone; thus, stabilin-1 is considered a marker of M2 polarization [85]. Stabilin-1 expression in TAMs supports tumor growth in animal models and correlates with poor prognosis for patients with various cancers [82,86,87]. Some ligands known to interact with stabilin-1 extracellular domain include modified low-density lipoproteins (LDL), such as oxidized LDL (oxLDL) and acetylated LDL (acLDL), and advanced glycation end products (AGEs), secreted protein acidic and rich in cysteine (SPARC), placental lactogen, and epidermal growth factor (EGF) [84,88,89,90]. As an endocytic receptor, stabilin-1 engages the clathrin-dependent pathway to facilitate the intracellular trafficking of internalized ligands. This process involves sorting within endosomes, leading to either lysosomal degradation or storage in secretory vesicles [79,88,91]. Using GGA adaptors, stabilin-1 can also sort intracellular ligands from the biosynthetic pathways into the lysosomal secretory route in macrophages [90].

We identified GDF-15 as a directly interacting protein of stabilin-1 through yeast two-hybrid screening. This interaction was confirmed by affinity chromatography and endocytosis assays. Furthermore, we observed that impaired clearance of GDF-15 in STAB-1^−/−^-STAB-2^−/−^ mice contributed to severe glomerular fibrosis and mild perisinusoidal hepatic fibrosis [77]. However, the fate of GDF-15 after internalization by these receptors remains unclear.

In vitro studies have investigated the target cells of rGDF-15 primarily by assessing its impact on the transcriptome through RT-PCR. Table 3 summarizes the major effects of rGDF-15. Takenochi et al. showed that rGDF-15 supplementation increases fibroblast activation measured by increased secretion of alpha-smooth muscle actin (α-SMA). This effect was not mediated by the GFRAL/RET activation pathway [43]. Other myeloid cells strongly affected by rGDF-15 treatment are erythrocytes. rGDF-15 supplementation leads to decreased erythroid colony formation and decreased transcription of erythroid differentiation factors [52]. In contrast, Shah et al. showed that the intratracheal administration of rGDF-15 leads to an increased erythrocyte count in mice lacking GDF-15 [92]. In vitro rGDF-15 supplementation has also been linked to increased angiogenesis and increased VEGFA expression in glioblastoma cells [93].

**Table 3 ijms-25-07313-t003:** Effects of rGDF-15.

Effects	Evidence	Method	Target Cells	Conditions	GDF-15 Source	Concentration (ng/mL)	Reference
Increased M2 and decreased M1 polarization	Increased Arg-1 expression Decreased IL-6, TNF-α, MCP-1, and IL-10 secretion, decreased CD80, increased CD163 mRNA levels	RT-PCR, Flow cytometry	THP-1 and RAW264.7	rGDF15 for 48 h	CHO; *E. coli*	100	[43]
Fibroblast activation	Increased α-SMA protein expression	Western blot	WI-38	Preincubation for 48 h; rGDF15 for 72 h	CHO; *E. coli*	0–100	[43]
	Increased Smad2/3 phosphorylation through TGF-β I receptor	Western blot	WI-38	Preincubation for 48 h; rGDF15 for 20 min-1 h	CHO; *E. coli*	100	[43]
Reduced metabolic activity in erythroid cells	Decreased optical density with increasing GDF-15 concentration	RT-PCR, flow cytometry, MTT cell metabolic activity assay	K562	Treatment with β-thalassemia serum containing 48 ng/mL of GDF-15 or GDF-15 for 24 h, 48 h and 72 h	CHO	2–50	[52]
Reduced proliferation in erythroid cells	Decreased optical density with increasing GDF-15 concentration in MTT assay	CFSE proliferation assay	K562	GDF-15 for 5 d	CHO	10–50	[52]
Increased angiogenesis	Increased VEGFA expression	Tube formation assay, RT-PCR	U373, HBMVEC	3 d in culture	CHO	100	[93]

Arg-1: Arginase-1; CFSE: carboxyfluorescein succinimidyl ester; CHO: Chinese hamster ovary; GDF-15: growth differentiation factor 15; HBMVEC: human brain microvascular endothelial cells; K562: lymphoblast; MCP-1: Monocyte Chemoattractant Protein-1; RT-PCR: reverse transcription real-time polymerase chain reaction; TGF-β: transforming growth factor-beta; THP-1: human acute monocytic leukemia cell line; TNF-α: tumor necrosis factor α; U373: glioblastoma cells; VEGF: vascular endothelial growth factor; WI-38: fibroblasts from lung tissue.

## 3. GDF-15 in Health and Pathology

GDF-15 basal plasmatic levels range from 337 to 1060 pg/mL. These levels may increase during certain physiological changes, such as muscle contraction and exercise [35,51], and under the pathological conditions reviewed in the following sections. Circulating GDF-15 increases with age and is unaffected by gender [35,42,94]. Notably, GDF-15 is significantly elevated at birth and during the first 4 months of life, reaching around three times the average adult concentration [95]. Its levels also progressively rise during pregnancy, correlating with gestational week and peaking in the third trimester [31].

Under physiological conditions, GDF-15 expression positively correlates with higher maturation states in the erythroid lineage [52]. Moon et al. found that rGDF-15 enhances the regulatory effect of Treg cells on activated T cells. Moreover, rs7226, a single nucleotide polymorphism associated with increased production of GDF-15 in humans, correlates with an increased count of lymphocytes and monocytes and decreased concentrations of innate immune cells and granulocytes [71].

As reviewed above, GDF-15 is overexpressed in cellular stress states and its function seems to be protective. Several pathological conditions show elevated plasma GDF-15 levels, including metabolic, cardiovascular, and hematological diseases and cancer, reaching concentrations up to one hundred times the physiological value (see Table 4) [94].

The most studied role of GDF-15 is its anorexigenic effect. GDF-15 mediates weight loss by reducing energy intake, an effect that is a direct consequence of GDF-15/GFRAL binding [75]. The mechanism behind this process was outlined in the previous sections. GDF-15 circulating levels have been consistently found to be elevated in obesity [106,107]. Pena-Leon et al. demonstrated in a mouse model that plasma levels of GDF-15 decrease during fasting and are restored after refeeding. This effect was associated with GDF-15 production by chief cells of the gastric mucosa. This group also showed that high GDF-15 in obesity is associated with increased post-transcriptional processing of the GDF-15 precursor, pro-GDF-15, which is highly accumulated in the gastric mucosa. Moreover, this post-transcriptional processing seems to be impaired in the fasting state [40].

GDF-15 supplementation improves the metabolic profile in high-fat diet-induced obese mice, showing a significant reduction in body weight, food intake, and glycemia. These effects may be due to taste aversion rather than increased gastric motility [106,107]. This anorexic effect is effectively abolished by blocking with a monoclonal antibody directed against GDF-15 and by GDF-15 or GFRAL knockout (KO) [69,106]. Recent studies have suggested that GDF-15’s effect on body weight may also be associated with increased energy expenditure and thermogenesis [108]. Recently, Feng Lu et al. found that GDF-15/GFRAL binding is an important determinant of the efficacy of ketogenic-diet-induced weight loss. They found that GDF-15 hepatic production significantly increases after introduction of ketogenic diet in obese pigs and mice, which was an effect of increased PPARγ, a known transcription factor of GDF-15. Additionally, they found that the ketogenic-diet-induced weight loss was mainly an effect of decreased energy intake, which corroborates previous studies highlighting the anorexigenic role of GDF-15 [109]. A recent review summarizes the association between GDF-15 and body weight [110].

High plasmatic GDF-15 levels are also found in association with impaired glucose tolerance, insulin resistance, diabetes, and diabetes-related complications, but its role in glucose homeostasis is unclear. Other references provide a detailed review of this matter [111,112]. GDF-15 is not only elevated during pregnancy but has also shown diagnostic relevance in preeclampsia and gestational diabetes mellitus [113,114].

### 3.1. GDF-15–Macrophage Interaction in Physiological and Pathological Conditions

In murine and human BMDM, as well as in THP-1 and RAW264.7 cells, GDF-15 expression is increased under stimulation of pro- and anti-inflammatory mediators, including IL-4, IL-1β, TNF-α, IL-2 and macrophage colony-stimulating factor (M-CSF) [48,54]. At the same time, GDF-15 has been widely associated with M2 differentiation and with the inhibition of M1 polarization [43,54,57]. Pence et al. examined the associaton between human serum GDF-15 levels and different parameters of monocyte immunosenescence. They observed significantly elevated levels of GDF-15 in the elderly population compared to younger individuals. Also, circulating GDF-15 levels displayed a negative correlation with maximal monocyte respiratory capacity [115]. The particular interplay between GDF-15 and macrophages in various pathological contexts will be reviewed in the following sections.

#### 3.1.1. Sepsis and Inflammation

Serum GDF-15 levels have been found to be elevated in sepsis, regardless of the pathogen involved, and have been correlated with prognosis, severity, and survival, as well as being elevated in SARS-CoV-2 in adults and children [54,116,117,118,119,120,121]. Assessment of the diagnostic value of GDF-15 highlighted its role as a biomarker of sepsis severity, including septic shock. GDF-15 showed a positive correlation with procalcitonin, IL-6, and IL-10 [54]. GDF-15 KO mice were protected against cecal ligation- and puncture-induced sepsis, showing less severe symptoms, lower markers of inflammation, and lower bacterial load [118]. Several other studies have proposed GDF-15 as a protective factor in sepsis [33,122,123]. High GDF-15 levels during bacterial inflammation stimulate beta-adrenergic sympathetic outflow and hepatic triglyceride production, mediate cardiac protection, and improve thermal homeostasis [123]. Supporting this mechanism, Kim et al. found increased catecholamine synthesis-related gene expression and increased thermogenesis in mice overexpressing GDF-15 [44]. Recently, Wang et al. also proved that GDF-15-treated mice show increased skeletal muscle noradrenaline and oxygen consumption, which is associated with increased maintenance of energy expenditure [124].

In vitro stimulation with LPS induces a dose- and time-dependent increase in both GDF-15 expression and secretion in THP-1 and RAW264.7 macrophages [54]. Similarly, in vivo LPS injection increases plasmatic GDF-15 concentrations in mice, rats, and humans (in humans, LPS was administered at a dose of 1 ng/kg) [33,125]. Pretreatment with rGDF-15 resulted in a dose-dependent decrease in the expression of proinflammatory cytokines such as TNF-α, IL-6, MCP-1, and IL-10. GDF-15 also appears to improve the phagocytic and bactericidal function of macrophages in the THP-1 and RAW264.7 cell lines [54]. Similar experiments by Govaere et al. showed decreased TNF-α and CCL2 secretion after treatment with rGDF-15 in THP-1 cells challenged with LPS [126]. Possible molecular pathways responsible for the decreased secretion of proinflammatory cytokines under GDF-15 exposure have been considered. Zhang et al. found that rGDF-15 promoted PI3K/Akt phosphorylation in macrophages under LPS-induced inflammation. This effect was reduced by treatment with the PI3K/Akt inhibitor [120]. Other research groups proposed the decrease in phosphorylation of JAK1/STAT3 and nuclear translocation of NF-κB after rGDF-15 treatment as cascade pathways [122]. The inhibitory effects of GDF-15 on NF-κB have also been shown by others [66,127]. In line with these findings, a novel mechanism for the effect of GDF-15 in sepsis was proposed. This mechanism suggests that GDF-15 inhibits glycolysis through MAPKs/NF-κB signaling inhibition in alveolar macrophages during sepsis conditions in vitro. The loss of glycolytic activity in alveolar macrophages contributes to a less inflammatory phenotype [128]. However, no specific receptor has been proposed to mediate the above-mentioned effects.

Although the anti-inflammatory effect of GDF-15 in macrophages has been known since its discovery, the direction of the systemic effects remains inconclusive. However, recent evidence has deepened the knowledge of the mechanism of GDF-15 in sepsis, and this has highlighted an exciting opportunity to redirect the therapy of such a complex disease.

#### 3.1.2. Fibrosis

GDF-15 has been linked to fibrotic diseases. Table 5 summarizes recent mouse model experiments in the context of fibrosis. Govaere et al. and Chung et al. found that GDF-15 expression in hepatic tissue positively correlated with fibrosis progression in NAFLD and CCl4-induced liver injury, respectively [126,129]. In contrast, Li et al. observed decreased GDF-15 expression levels in mouse and human liver fibrotic tissue compared to healthy liver tissue during fibrosis progression [130]. These research groups showed that GDF-15-deficient mice display increased liver fibrosis, which can be mitigated with rGDF-15 treatment. Chung et al. also demonstrated that mitochondrial damage, caused by oligomycin and rotenone, induces GDF-15 expression. In a co-culture model involving hepatocytes and Kupffer cells, they observed that the absence of GDF-15 production by hepatocytes resulted in increased expression of proinflammatory cytokines following LPS stimulation by Kupffer cells. GDF-15 deficiency altered immune infiltration in hepatic tissue, increasing the number of CD4+, CD8+, T cells, and neutrophils, while leaving the monocyte population unaffected. This effect was reversed through the supplementation with rGDF-15 [129]. These findings highlight the paracrine effects of GDF-15 in fibrosis progression.

Using a GDF-15-deficient mouse model, Li et al. demonstrated an increase in macro-phage infiltration in liver tissue compared to non-deficient mice [130]. Moreover, the predominant macrophage infiltrating phenotype was shifted from Ly6Clow to Ly6Chi, and NF-κB signaling was hyperactivated in GDF-15 KO mice, while rGDF-15 was able to alleviate hepatic fibrosis. The authors suggested that this may be a consequence of decreased M1 polarization and reduced expression of proinflammatory cytokines in the liver in the presence of GDF-15. However, this study did not show how GDF-15 exerts these effects in the macrophage system, whether through a peripheral receptor in a paracrine manner or through a macrophage receptor itself. Finally, Li et al. tested the effect of GDF-15 preprogrammed macrophages administered parenterally in mice with CCl4-induced liver fibrosis. This cell therapy successfully decreased the severity of hepatic fibrosis [130].

Kupffer cells are essential contributors to the progression of alcohol-induced liver damage, leading to its fibro-inflammatory stages. Kim et al. suggested that increased catecholamines play a protective role in preventing the progression of alcohol-induced hepatic fibrosis in chronic ethanol-fed mice [132]. They showed that catecholamines can induce apoptosis in Kupffer cells under ethanol-induced oxidative stress. This effect was shown to be mediated by beta-1 adrenergic receptor 1/2 (ADRB1/2), a catecholamine receptor whose expression in Kupffer cells was upregulated by GDF-15. They also demonstrated high levels of GDF-15 in the hepatic tissue in the alcohol-induced hepatic fibrosis model. The decrease in Kupffer cells prevented the accelerated fibrosis progression. These findings shed light on the mechanism by which GDF-15 functions as a stress-induced cytokine, promoting apoptosis of inflammatory Kupffer cells, thereby mitigating further hepatic damage [132].

GDF-15 also associates with lung fibrosis. GDF-15-deficient neonatal mice, which were briefly exposed to hyperoxia, presented decreased survival rates along with impaired alveolarization and perturbed macrophage activation in lung tissue [131]. In a bleomycin-induced lung fibrosis mouse model, GDF-15 expression and protein levels are increased in the lung tissue, bronchoalveolar fluid, and plasma of mice with pulmonary fibrosis. Within the lung tissue, the highest GDF-15 positivity was found in epithelial cells and macrophages [43].

#### 3.1.3. Regenerative Processes

Titanium is a widely used implant material in fields such as orthopedics, cardiology, and dentistry [134]. Our group has shown that macrophages exposed to titanium nanoparticles (TiNPs) increased their expression and secretion of GDF-15 [135]. Siddiqui et al. investigated the role of GDF-15 in prostate cancer bone metastasis. They found that prostate cancer cells highly express and secrete GDF-15, which further induces the expression of osteoclastogenesis-related genes in osteoclasts and the expression of chemoattractant protein-1 (MCP-1/CCL2), which is involved in macrophage recruitment to osteoblasts in mice [61]. This suggests that GDF-15 may activate osteoclastogenesis through a paracrine mechanism and further recruit macrophages in the bone. Furthermore, this group found the presence of GFRAL/RET in osteoblasts and showed that GFRAL silencing decreases osteoclastogenesis and macrophage recruitment markers’ expression, induced by GDF-15 [61]. This proposes a novel GDF-15/GFRAL/RET functional interaction. We also found that TiNPs decrease the expression of stabilin-1, the clearance receptor of GDF-15, in macrophages and decrease their endocytic function [135]. This mechanism could further increase the local levels of GDF-15 surrounding the titanium implant microenvironment and contribute to inadequate implant osseointegration and aseptic loosening.

#### 3.1.4. Cancer

GDF-15 has been recognized as a potential diagnostic and prognostic biomarker for several gastrointestinal tumors, including pancreatic, colorectal, esophageal, hepatocellular, and gastric cancers [111]. In addition, other cancers such as glioblastoma, breast, lung, cervical, ovarian, endometrial, lung, prostate, renal, urothelial, thyroid, and melanoma have also shown elevated levels of the cytokine [93,104,112,136].

GDF-15 appears to have a dualistic function in the process of carcinogenesis. It has an inhibitory effect on tumor growth in the early stages and subsequently facilitates progression and metastasis in the advanced stages, as shown in Figure 3 [137]. However, contradictory associations have been uncovered in various tumor models and cancer types.

Concerning primary tumor growth, rGDF-15 has been shown to promote proliferation in esophageal carcinoma cells. Also, high levels of GDF-15 in the tumor microenvironment, produced by both esophageal cancer cells and macrophages, are associated with more malignant phenotypes [138]. An autocrine mechanism has been identified in pancreatic ductal carcinoma cells, where pancreatic cells were shown to both secrete GDF-15 and express its receptor GFRAL. An increased GFRAL expression promotes tumor growth in a dose-dependent fashion with increasing GDF-15 concentration. Tumor growth was concordantly decreased by GDF-15 knock down (KD) [139]. However, no GFRAL KD or KO experiment was conducted, and the rGDF-15 concentrations utilized in the experiments were exceedingly high. This is the only tumor model in which the pathway GDF-15/GFRAL has been demonstrated. This mechanism was confirmed using xenograft mouse models and suggests a potential therapeutic approach for solid tumors. For instance, the use of GDF-15 to shift from cold to hot tumors may increase the susceptibility to chemotherapy.

In contrast, overexpression of GDF-15 has been shown to decrease cell proliferation and invasion in metastatic bladder cancer cell lines [38]. However, the rGDF-15 concentrations used in these experiments were considerably higher than those secreted by the cancer cells. They did, nevertheless, demonstrate the same finding in the mouse model when performing KD experiments. It is worth noting that they examined several bladder cancer cell lines, including the papilloma bladder cancer cell line, in which GDF-15 is expressed at higher levels compared to the metastatic cell lines. As a possible mechanism, they suggested that methylation of the GDF-15 gene, which leads to its lower levels in bladder cancer, may contribute to tumor progression. They showed that GDF15 knockdown decreased F-actin polarization and thereby tumor invasion [38].

Another cancer type in which GDF-15 has been associated with a more favorable outcome is renal cell carcinoma (RCC). Patients with RCC and elevated GDF-15 levels in tumor tissue exhibit a more favorable prognosis compared to those with lower levels. Yang et al. demonstrated that GDF-15 induces ferroptosis by upregulating GPX4, a mechanism that could explain its inhibitory role in RCC progression [140].

Likewise, patients with renal cell carcinoma (RCC) and increased GDF-15 protein levels in tumor tissue show a better outcome. Yang et al. proposed GDF-15 as a possible regulator of ferroptosis in RCC [140]. However, the molecular mechanism supporting this hypothesis needs to be investigated. Additionally, the aforementioned experimental work focused exclusively on the cancer line, bladder or renal, without considering the role of the tumor microenvironment. Yang et al. did not utilize an animal model to support the proposed role of GDF-15 in RCC. In these group of experiments, the tumor progression was only evaluated in early stages and in in vitro models.

Another hallmark of cancer influencing tumor growth is angiogenesis. GDF-15 has been shown to activate the hypoxia-inducible factor-1α (HIF-1α). HIF-1α is a key regulator of VEGF, an angiogenic factor that promotes endothelial cell proliferation, migration, and new blood vessel formation, thus facilitating tumor angiogenesis. HIF-1α/VEGF signaling pathway activation in the presence of GDF-15 has been evidenced in colon, gastric, and breast cancer cells [141,142]. This angiogenesis pathway was shown to be induced through the transactivation of ErbB2 in gastric and breast cancer cells [142]. Additionally, rGDF-15 also promotes p53 degradation and increases HIF-1α accumulation and vessel formation in human umbilical vein endothelial cells [143]. The intracellular cascade activation mechanism surrounding GDF-15 was not demonstrated, nor was the receptor responsible for this effect.

Recently, Chitinase-3-like protein 1 (YKL-40), produced by macrophages, was shown to promote GDF-15 expression in gallbladder tumor cells. The interaction of YKL-40 and GDF-15 has been demonstrated to promote tumor invasion and to upregulate PD-L1 expression in tumor cells. PD-L1 binds to its receptor PD-1 on T cells, resulting in the suppression of anti-tumor activity of CD8+ T lymphocytes, thereby contributing to immune evasion [144]. Furthermore, the induced expression of PD-L1 by GDF-15 has been demonstrated in glioblastoma cells. Nevertheless, the involvement of macrophages was not investigated in this case [145]. Although YKL-40 was demonstrated to be expressed by M2 macrophages in the tumor microenvironment in a single-cell analysis by Wang et al., for further validation, YKL-40 was derived from THP-1 macrophages, which are not primary human cells. Nevertheless, the cooperation between YKL-40 and GDF-15 provides a compelling illustration of the significance of the interplay between macrophages within the tumor microenvironment [144]. Supporting this observation, patients with low plasmatic levels of GDF-15 show better response rates to anti-PD-1/PD-L1 inhibitors in advanced non-small cell lung cancer and in melanoma [146,147]. Recently, Haake et al. reported that GDF-15 produced by melanoma cells inhibits lymphocyte adhesion to endothelium and migration through inhibition of the lymphocyte function-associated antigen (LFA-1)/intercellular adhesion molecule 1 (ICAM-1) axis in T lymphocytes. This inhibition results in reduced lymphocyte infiltration at the tumor site when GDF-15 tissue levels are elevated. In addition, combined anti-GDF-15 and anti-PD-1 therapy results in increased T lymphocyte infiltration in mouse models of pancreatic cancer [147]. The use of GDF-15 as a marker for patients who would benefit from such therapy may be a valuable option in such cancers, where anti-PD-1/PD-L1 inhibitors are used as first-line therapy in intermediate- and poor-risk metastatic tumors.

Other authors linked GDF-15 to chemotherapy resistance, as in the case of Zheng et al., who showed that TAMs derived from a mouse xenograft model of colorectal cancer secrete high levels of GDF-15 and contribute to the reduced chemosensitivity in colorectal cancer cells by increasing fatty acid oxidation metabolism [148]. A similar mechanism has been suggested by Yu et al. in gastric cancer [149].

The paradoxical involvement of GDF-15 in carcinogenesis can be explained by a corresponding shift in macrophage phenotype. In the early stages of cancer, the presence of GDF-15 in the tumor microenvironment would reduce the M1 phenotype and thus the immune surveillance, favoring tumor growth [66]. Late in the process, high levels of GDF-15 increase M2 infiltration, favoring angiogenesis, metastasis, and chemotherapy resistance [149]. Bonaterra et al. showed that the presence of GDF-15 in prostate cancer tissue was associated with macrophage infiltration, and the presence of GDF-15+ macrophages was associated with high-grade malignancy [150]. This was also highlighted by Sadasivan et al., who found a higher risk of biochemical recurrence in patients whose prostate cancer biopsy was enriched for M2 macrophages and characterized by elevated GDF-15 expression [151].

Lv et al. studied the effect of GDF-15-enriched conditioned medium from M1-polarized THP-1 macrophages on SCC25, a tongue squamous cell carcinoma cell line, and showed that phosphorylation of ErbB2 and its signaling proteins ERK and AKT was increased. This effect was reduced by knocking out GDF-15 in SCC25 cells [152]. This activation pattern has been widely implicated in tumor progression in several cancer types, particularly breast cancer [144,153]. These observations again highlight a tumorigenic effect of GDF-15. However, a direct correspondence between the presence of GDF-15 in the tumor microenvironment and squamous cell carcinoma progression has not been established [152]. Ratnam et al. highlighted the interaction between NF-κB and GDF-15 by showing that constitutive activation of NF-κB in pancreatic cancer cells leads to secretion of GDF-15, a known NF-κB inhibitor, and induces a decreased cytotoxic capacity in TAMs [66].

Elevated GDF-15 has been strongly associated with metastasis in prostate, esophageal, hepatocellular, colorectal, pancreatic, gastric, and endometrial cancers [154]. Esophageal, breast, and colon cancer models show that GDF-15 correlates with the loss of E-cadherin and that the inhibition of GDF-15 expression decreases cell migration and invasion ability [73,155,156]. In contrast, in A549 lung cancer cells, overexpression of GDF-15 reduces cell growth and migration and decreases the spread of lung and bone metastases [157]. The exact effects and mechanisms explaining GDF-15 behavior in cancer are still controversial and often paradoxical. Regarding the interplay between GDF-15 and macrophages in metastasis, Ding et al. showed that GDF-15 secreted by macrophages contributes to an invasive phenotype in colon cancer cells, an effect that was reversed by GDF-15-neutralizing antibodies. The proposed mechanism was increased phosphorylation of c-Fos via Erk1/2 activation by GDF-15, which induced the expression of epithelial–mesenchymal transition in colon cancer cells [158].

The precise role of GDF-15 in cancer remains unclear. Nevertheless, further experiments should always consider the interplay within the tumor microenvironment. Moreover, it is of interest to determine whether the observed effects in cancer are independent of GFRAL presence. New immunochemistry evidence suggests that GFRAL is not exclusively localized in the midbrain in murine models but rather is also found in hepatocytes, visceral fat tissue, proximal tubule cells in the kidney, enterocytes in the intestine, and in cardiac and skeletal muscular structures [159]. Zhao et al. reported the presence of GFRAL in pancreatic cancer, as well as in hepatocellular carcinoma, cholangiocarcinoma, colorectal carcinoma, and RCC. The latter demonstrated a reduction in GFRAL expression [139]. However, the direct association between GDF-15 and GFRAL was only demonstrated in pancreatic cancer. This finding opens the door for the validation of further interactions of GDF-15 and GFRAL in the aforementioned cancer types. If confirmed, this may be a possible explanation for the contradictory growth-promoting effects evidenced in several cancer types, but not in RCC, in which GFRAL presence is decreased and in which GDF-15 rather inhibits progression. Furthermore, the potential paracrine effect of macrophages producing GDF-15 in the tumor microenvironment expressing GFRAL should also be explored.

#### 3.1.5. Metabolic and Cardiovascular Disorders

Besides its anorexigenic effect and its effect on glucose metabolism, GDF-15 has also been proposed as a biomarker for increased mortality risk and recurrent myocardial infarction (MI) after acute coronary syndrome. Similarly, GDF-15 serves as a biomarker in heart failure, a common complication in patients with coronary heart disease and in atrial fibrillation [105,160,161,162]. In fact, GDF-15 is positively correlated with cardiovascular mortality and all-cause mortality [163]. Mice deficient in GDF-15 have higher mortality after induced MI. They also display an increased recruitment of polymorphonuclear leukocytes, monocytes, and macrophages in the myocardial tissue as compared to controls. Additionally, this recruitment is reduced upon treatment with rGDF-15, which decreases leukocyte adhesion, arrest, and transmigration on the endothelium [164]. Taken together, elevated GDF-15 levels after MI may exert a protective function by reducing immune cell recruitment and, thereby, MI complications, such as cardiac remodeling and heart failure.

Recently, Xiao et al. explored the programming of myocardial microenvironment using extracellular vesicles (EVs) containing GDF-15. They used EVs derived from M2 transfected with GDF15. In a rat model, intracardiac administration of EVs showed a significant reduction in the development of cardiac fibrosis and remodeling after MI compared to control. EV treatment also induced angiogenesis in the injured myocardium. Like others, the authors attributed these findings to the increased polarization of the M2 phenotype in the myocardium of EV-treated rats. Interestingly, the authors provide a mechanism by which GDF-15 mediates FABP4 inhibition through Smad2/3, resulting in M2 polarization [165]. This finding is consistent with Takenouchi et al. and Li et al., who demonstrated the phosphorylation of Smad2/3 following GDF-15 binding to the TGF-βI receptor [43,72]. Nevertheless, how FABP4 inhibition further leads to M2 polarization was not addressed. The study by Xiao et al. also highlights EVs as a possible endogenous vehicle for the peripheral effects of GDF-15, for which no acceptable receptor has yet been recognized [165]. Although the beneficial effects of EVs containing GDF-15 were significantly better than those of EVs without GDF-15, EVs lacking GDF-15 also contributed to myocardial repair compared to PBS treatment. This suggest that there are more components in the M2-derived EVs that improved myocardial injury than just GDF-15. Since this study did not include the pure injection of GDF-15 into the myocardial tissue as a comparison group, the pure effect of GDF-15 as a possible therapy cannot be concluded [165].

Taken together, elevated GDF-15 levels after MI may exert a protective function by reducing proinflammatory immune cell recruitment and, thereby, MI complications, such as cardiac remodeling and heart failure. The addition of GDF-15 to the injured myocardium may further accelerate the healing process and novel approaches to this aim should be explored. Xiao et al. elegantly showed an alternative delivery method using EVs as a vehicle for GDF-15, which is an interesting cell-free therapy [165]. However, in the context of MI, using intracardiac injection to ensure the adequate distribution of EVs is still a very invasive approach.

In the context of allogeneic transplantation of cardiac progenitor cells (CPCs) as a therapy for myocardial infarction, downregulation of GDF-15 in CPCs resulted in decreased activation of Tregs and M2 macrophages, preventing an adequate engraftment into the injured myocardium [133]. In contrast, GDF-15 secreted by CPCs inhibited NF-κB activation and promoted a shift toward M2 polarization and Tregs activation, ultimately associated with a cardioprotective outcome. In this series of experiments, the effects of GDF-15 as part of the secretome of CPCs were shown to be mediated by Tregs, as CPCs injected in absence of Tregs failed to promote the protective effects [133].

GDF-15 has emerged as a significant player in mechanisms involved in atherosclerosis and macrophage function. For example, treatment of THP-1 with rGDF-15 is associated with lipid accumulation, whereas GDF-15 knockdown resulted in reduced lipid burden. In addition, rGDF-15 increased the levels of autophagy-related proteins, suggesting a possible role for GDF-15 in autophagosome formation in foam cells [166]. Heduschke et al. supported these findings by showing that siGDF-15 decreased the autophagic activity in THP-1, an effect that was reversed by rGDF-15 supplementation in THP-1 cells [167]. The observation that GDF-15 decreases the release of proinflammatory cytokines has been reported to be associated with the decreased expression of TLR4 in macrophages under oxLDL treatment [168]. This suggests a potential immunomodulatory role of GDF-15 in the progression of atherosclerosis by reducing the proinflammatory plaque surroundings. In contrast, Bonaterra et al. showed that GDF-15 KO in ApoE^−/−^ mice exhibited a reduced degree of endothelial stenosis. This study highlights a beneficial phenotype in the absence of GDF-15 [169]. Nevertheless, analysis of the plaque surrounding and characterization of the macrophage phenotype were not performed.

The precise mechanism by which GDF-15 contributes to the interaction between macrophages and atherosclerosis remains uncertain. Further research should include models that integrate the interplay between macrophages, vascular smooth muscle cells, and endothelial cells in the development of atherosclerotic plaques. Additionally, the use of THP-1 cells may not be the most optimal model, as macrophages found in atherosclerotic lesions are also of tissue origin [170].

## 4. Conclusions

The reviewed studies emphasize the significant interplay of GDF-15 in the regulation of immune responses and paracrine effects involving the macrophage system. Although the interplay between GDF-15 and the macrophage system has been identified since the discovery of GDF-15, there is still a knowledge gap regarding the extent to which this interaction impacts human pathology as well as in the understanding of the dynamic mechanism of GDF-15 production and responses of the target cells. In order to understand the sequence of events in GDF-15-controlled processes, further experimentation is needed and the mechanisms of target cell responses to GDF-15 have to be deciphered. One notable gap is the lack of identification of macrophage receptors for GDF-15 that can clarify its immunomodulatory effects.

## 5. Future Directions

Due to its crucial roles in multiple biological processes, GDF-15 has been identified as a potential therapeutic target for various diseases. Currently, there is an ongoing recruitment for Phase I/II clinical trials investigating the impact of neutralizing GDF-15. Visugromab is an anti-GDF-15 monoclonal antibody currently being evaluated in a Phase II clinical trial as a combination therapy with the checkpoint inhibitor anti-PD-1/PD-L1 for treating advanced solid tumors. AV-380 and Ponsegromab, also anti-GDF-15 antibodies, are under evaluation as potential therapies for cancer-induced cachexia in non-small cell lung, pancreatic, and colorectal cancer patients, as well as in metastatic colorectal cancer, through Phase I and II studies. The aim of this therapeutic approach is to prevent the binding of GDF-15 to GFRAL, thereby improving anorexia and cachexia. The Phase Ib trial included 10 participants receiving Ponsegromab, with no serious adverse effects. Additionally, a gain of approximately 4,6 kg compared to baseline was achieved in 12 weeks [171]. Although the number of participants is insufficient to draw definitive conclusions, this antibody therapy represents a promising alternative for cachexia. Furthermore, the inclusion criteria include patients with the aforementioned cancer types, which may facilitate the human study of the effect of GDF-15 inhibition on tumor growth. The Phase I/II study is currently evaluating NGM120, a GFRAL antagonist antibody, on cachexia in participants with advanced pancreatic and prostate cancer.

Additionally, a current application for monoclonal anti-GDF-15 is in the treatment of heart failure, with Ponsegromab undergoing recruitment in a Phase II study. Finally, a multicenter Phase II study on Visugromab is recruiting participants to assess the effects of combination therapy with Nivolumab on muscle-invasive bladder carcinoma. The Phase I/IIa clinical trial of AZD8853 in patients with metastatic solid tumors was terminated prematurely due to an overall evaluation of its risk–benefit profile.

Overall, targeting GDF-15 is a very appealing therapeutic field for those pathologies where GDF-15 activity has demonstrated a clear impact, such as fibrosis-associated diseases. Conversely, animal studies on GDF-15 modulation in sepsis and cancer display significant discrepancies. Consequently, a comprehensive investigation of tissue-specific and context-dependent effects is necessary to clarify these inconsistencies.

Further research should focus on deciphering the complex molecular mechanisms governing the actions of GDF-15, exploring its potential as a therapeutic target, and elucidating the contextual factors that impact its expression in diverse conditions. In addition, exploring the role of cell-specific receptors in mediating the effects of GDF-15 presents an exciting opportunity for further investigation. A thorough comprehension of the role of GDF-15 in fibrosis may facilitate the development of innovative therapeutic interventions and improve the clinical management of fibrotic diseases, as well as prevent the possible adverse effects of GDF-15-based therapies.

As reviewed here, the new findings on GDF-15 in the macrophage system reveal a significant potential for displaying immunomodulatory properties in contexts involving remodeling, such as MI and fibrosis. For example, macrophages expressing GDF-15 have been suggested as a novel macrophage type with distinct transcriptomics, which mediates remodeling in sterile muscle injury, specifically by enhancing myoblast proliferation and decreasing inflammatory infiltration [63]. Further exploring the potential benefits of GDF-15+macrophages as an alternative to GDF-15-treated macrophages or rGDF-15 alone is very appealing. Dai et al. recently identified a naturally occurring cluster of GDF15^high^ macrophages in the human lung. These macrophages exhibit distinct characteristics from M1 and M2 macrophages and express high levels of GDF-15. Additionally, they display a tolerogenic phenotype in vitro [172]. A more thorough investigation of the characteristics of this cluster is necessary to ascertain its suitability as a candidate for cell therapy with respect to functionality, plasticity, longevity, and immunogenicity. Figure 4 summarizes the reported cellular effects of GDF-15 in the macrophage system as well as its systemic interactions.

## Figures and Tables

**Figure 1 ijms-25-07313-f001:**
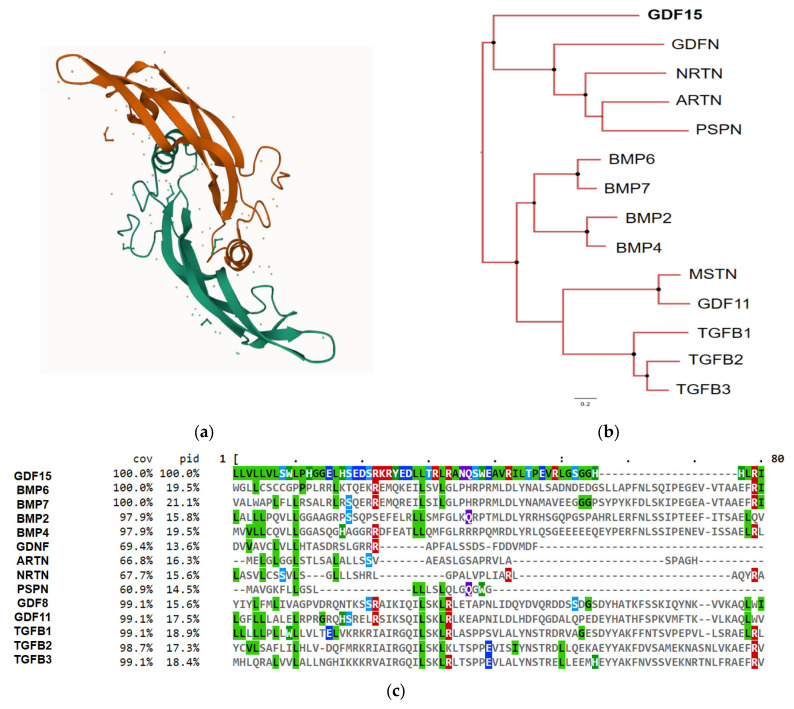
GDF-15 crystal protein structure and phylogenetic associations. (**a**) Crystal structure of the GDF-15 protein. Obtained from InterPro: https://www.ebi.ac.uk/interpro/structure/PDB/5vt2/#table (accessed on 22 September 2022) [30]; (**b**) phylogenetic tree illustrating relationships among GDF-15, TGF-β, and GDNF superfamily members. Generated using TaxOnTree:(bioinfo.icb.ufmg.br/taxontree/#x) (accessed on 4 August 2023); (**c**) protein sequence alignment of GDF-15 compared to TGF-β and GDNF superfamily members. Sequence alignment was performed using Protein BLAST: (blast.ncbi.nlm.nih.gov) (accessed on 4 August 2023). cov: percentage of coverage, pid: percentage of identity.

**Figure 2 ijms-25-07313-f002:**
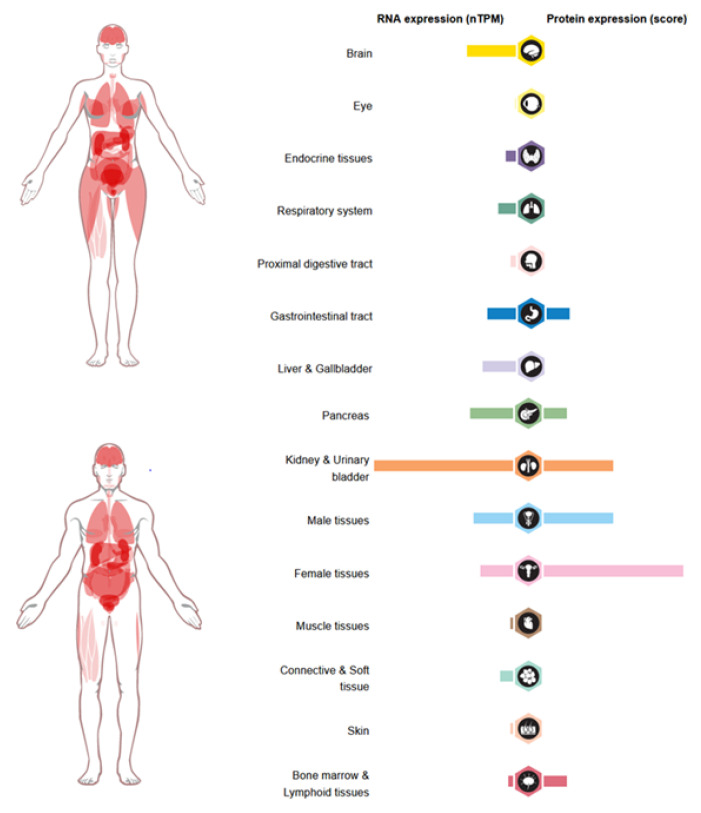
GDF-15 RNA and protein expression in the human body. GDF-15 expression is elevated in various tissues, including kidney, bladder, brain, and female and male tissues. RNA expression levels are represented using normalized transcript expression values (nTPM). Adapted from: Human Protein Atlas https://www.proteinatlas.org/ENSG00000130513-GDF15/tissue (accessed on 9 August 2023).

**Figure 3 ijms-25-07313-f003:**
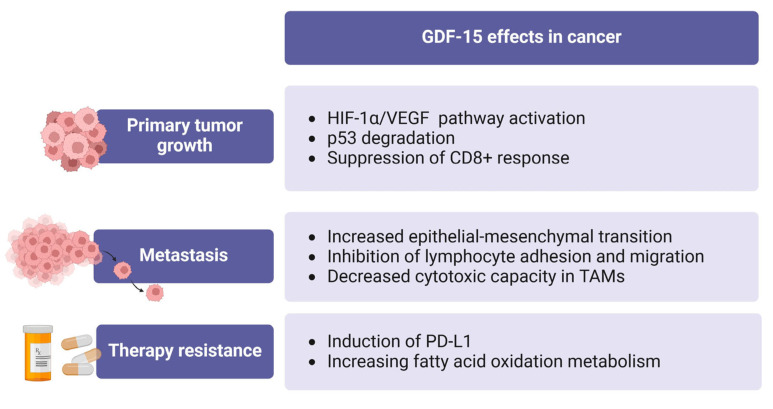
Effects of GDF-15 in carcinogenesis. Image created by biorender.com (accessed on 24 June 2024).

**Figure 4 ijms-25-07313-f004:**
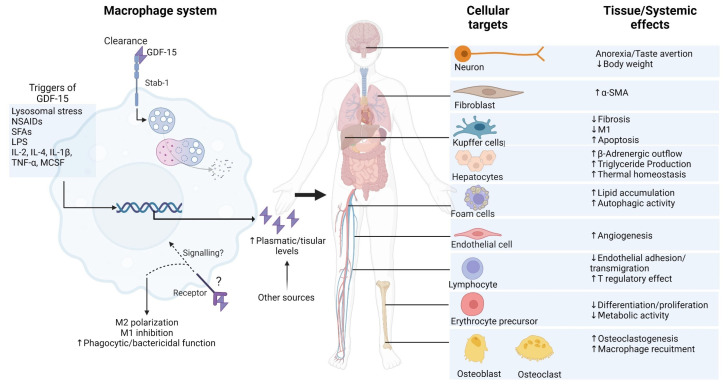
Overview of the effects of GDF-15 in the macrophage system. Image created by biorender.com (accessed on 18 June 2024).

**Table 4 ijms-25-07313-t004:** GDF-15 serum levels in pathology.

Condition	Concentration (pg/mL)	Reference
α-thalassemia syndrome	5900 ± 1200	[46]
Benign prostate hyperplasia	731 ± 500	[96]
β-thalassemia major	66,000 ± 9600	[46]
Chronic pancreatitis	2368 ± 2431	[97]
Colorectal carcinoma	783 ± 491	[98]
SARS-CoV-2 in ICU	12,400	[99]
Endometrial cancer	1077	[100]
Exercise	200–1000	[35]
Heart failure	2705	[101]
Hereditary hemochromatosis	720 ± 50	[46]
Mitochondrial myopathy	2711 ± 2459	[102]
Ovarian cancer	96.1–1876	[102]
Pancreatic cancer	5388 ± 3720	[97]
Preeclampsia	421 ± 187	[103]
Pregnancy	6300–15,300	[31]
Prostate cancer	860 ± 850	[96]
Renal cell carcinoma	1100 ± 150	[104]
Sepsis	4900	[54]
Sickle cell anemia	880 ± 160	[46]
Smoking	1835	[50]
Stable coronary heart disease	915–1827	[105]
Systemic sclerosis	1367	[50]

GDF-15: growth differentiation factor 15; ICU: intensive care unit.

**Table 5 ijms-25-07313-t005:** GDF-15 effects in fibrosis mice models.

Setting	Intervention	Finding	Reference
CCl4-induced liver fibrosis	CRISPR-Cas GDF-15 KO	Histopathology (H&E and Sirius red staining): increased fibrosis. Immunohistochemistry: increased macrophage F4/80 and neutrophil MPO infiltration, upregulated Ly6Chi Serum: increased hepatic enzymes	[130]
DDC-induced liver fibrosis	CRISPR-Cas GDF-15 KO	Histopathology (H&E and Sirius red staining): increased fibrosis and collagen deposition	
CCl4-induced liver fibrosis	AAV8 gene vector GDF-15 overexpression	Histopathology (H&E and Sirius red staining): reduced liver injury and fibrosis. Serum: reduced hepatic enzyme levels in blood. mRNA levels (RT-PCR): reduced IL-1β, TNF-α, and NOS2. Increased YM1, Arg1, and CD206	
DDC-induced liver fibrosis	CRISPR-Cas GDF-15 KO	Histopathology (H&E and Sirius red staining): reduced liver injury and fibrosis	
CCl4-induced liver fibrosis	Tail vein infusion with GDF15-preprogrammed macrophages 24 h	Histopathology (H&E and Sirius red staining): reduced liver injury and fibrosis. Serum: reduced hepatic enzyme levels	
Hyperoxia 95% after birth	GDF15^−/−^ mice	Higher mortality and lower body weight. Immunofluorescence for von Willebrand factor: impaired alveolarization and lung vascular development, lower macrophage F4/80 infiltration	[131]
Ethanol-induced liver disease	Genetic ablation of hepatocyte-derived GDF-15	Annexin V apoptosis assay: decrease in Kupffer cell apoptosis in liver perivenous region	[132]
Ethanol-induced liver disease	GDF15 KO	Histopathology (H&E and Oil Red O staining): increased hepatic fat accumulation. Serum: increased hepatic enzymes and triglyceride. Increased TNF-α and IL-6	[129]
CCl4-induced liver fibrosis	GDF15 KO	Histopathology (H&E and Oil Red O staining): increased hepatic fat accumulation. Serum: increased hepatic enzymes and triglyceride. Increased TNF-α and IL-6	
CCl4-induced liver fibrosis GDF15 KO	rGDF-15 0.5 mg/kg i.v.	Histopathology (H&E and Oil Red O staining): reduced collagen accumulation. Western blot: inhibition of NF-κB, JNK, and p38 signaling pathways	
Coronary artery ligation-induced myocardial infarction	Allogenic cardiac progenitor cells transplant with GDF-15 KD	Flow cytometry of cell suspension: decrease in M2 phenotype and Treg activation	[133]
Sterile muscle injury with cardiotoxin injection	rGDF-15 i.m.	Flow cytometry: decreased CD45+ muscle infiltration. Increased MCHII expression by monocyte-derived macrophages, anti-inflammatory phenotype	[63]

DDC: 3,5-Diethoxycarbonyl-1,4-Dihydrocollidine; CCl4: carbon tetrachloride; GDF-15: growth differentiation factor 15; H&E: Hematoxylin-eosin, i.v.: intravenous; i.m.: intramuscular; KO: knock-out; KD: knock-down; NF-κB: nuclear factor kappa-light-chain enhancer of activated B cells; NOS2: nitric oxide synthase 2; TNF-α: tumor necrosis factor α.

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
