# Peer review of "Macrophages as a Source and Target of GDF-15"

_ijms, 2024, doi:10.3390/ijms25137313_

Round 1

Reviewer 1 Report

Comments and Suggestions for Authors

  Silva-Bermudez et al. aim to review the effect of GDF-15 on macrophages under physiological and pathological contexts in order to discuss its therapeutic potential in distinct diseases. Although the topic is interesting the review should be improved by a more critical discussion of GDF-15 in the different pathologies, rather than a poorly report of the results of the different studies. The review starts with a broad overview of GDF-15 biology, encompassing its molecular structure, expression in the human body and biological activity. Within this topic, macrophages emerge just as one of many sources and targets of this pleiotropic molecule.  Next, the review focused on GDF-15 and macrophages in physiology and in several pathological contexts. However, it sometimes looks like a just a simple list of distinct studies regarding the effect GDF-15 on macrophages and/or disease outcome rather than a critical analysis of the findings that came up from these studies. As a reader, I would rather the authors highlighted and discussed thoroughly discrepancies in the different chapters, where the results of the studies are described.  Additionally, I think that the authors should try to suggest potential explanations of such discrepancies (different models of liver fibrosis, different mouse strains, different approaches…) and eventually end up with a clear statement about the potential beneficial or detrimental role of GDF-15 or, in case of lack of significant results, of the need for future studies, at the end of each chapter. In contrast, I appreciated the discussion of gaps of knowledge in the “future directions” chapter and I found tables and figures very comprehensive and clear.  

Comments on the Quality of English Language

I think that proofreading by a native English speaker might greatly improve the quality of the review and its overall impact.

Many sentences need rephrasing to improve clarity and accuracy. I have reported a few examples below, but there are many others throughout the manuscript.  

·      Abstract “Macrophages serve as both producers and receptors of GDF-15” might be rephrased as “Macrophages are both a source and a target of GDF-15”. “Receptor” is a molecule not a cell

·      Lines 163-164 “In adipocytes, signal transducer and activator of transcription 6 (STAT6) triggers the expression of GDF-15 when exposed to IL-13 treatment.”  Might be re-phrased as “upon IL-13 treatment, activation of signal transducer and activator of transcription 6 (STAT6) triggers the expression of GDF-15 in adipocytes” or “In IL-13-treated adipocytes, signal transducer and activator of transcription 6 (STAT6) triggers the expression of GDF-15”. Indeed, adipocytes are exposed to IL-13, while the cytokine-induced signaling cascade leads to the activation of the transcription STAT6.

·      Lines 462 . “However, the observed effects were not explained due to the binding of GDF-15 to a peripheral receptor or a macrophage receptor. “ I do not understand this sentence, if the effect of GDF-15 on liver fibrosis is independent on the binding of GFD-15 to a receptor which is the potential mechanism driving GDF-15 biological activity?

·      Lines 463-464. “Additionally, Li et al. tested the systemic effect of parenteral GDF-15 reprogrammed macrophages in mice with CCl4-induced liver fibrosis”. “Systemic” is redundant because the GDF-15 is administrated by intravenous injection. Furthermore, I do not understand the meaning of the sentence. “Li et al. tested whether the adoptive transfer of GDF-15-preprogrammed macrophages in the blood circulation mitigate CCl4-induced liver fibrosis in mice” or “Li et al. tested the effect of parenterally administrated GDF-15 on macrophage reprogramming in a murine model of CCl4-induced liver fibrosis”?

·      Line 474. Similarly, “ADRB1/2 is a catecholamine receptor, which is stimulated by GDF-15” means that GDF-15 binds and activates ADRB1/2 receptor or that GDF-15 upregulates the ADRB1/2 receptor?

·      Line 523. “the specific pathway surrounding this hypothesis needs to be investigated.” The molecular mechanism supporting this hypothesis needs to be investigated”

·      Line 533. “The interaction of YKL-40 and GDF-15 leads to tumor invasion and suppression of the immune response of CD8+ T lymphocytes through the induction of PD-L1 in gallbladder tumor cells” might be re-phrased as “The interaction of YKL-40 and GDF-15 promotes tumor invasion and upregulates PD-L1 expression in gallbladder tumor cells, which results in the suppression of anti-tumor CD8+ T lymphocytes”.

Author Response

Silva-Bermudez et al. aim to review the effect of GDF-15 on macrophages under physiological and pathological contexts in order to discuss its therapeutic potential in distinct diseases. Although the topic is interesting the review should be improved by a more critical discussion of GDF-15 in the different pathologies, rather than a poorly report of the results of the different studies. The review starts with a broad overview of GDF-15 biology, encompassing its molecular structure, expression in the human body and biological activity. Within this topic, macrophages emerge just as one of many sources and targets of this pleiotropic molecule.  Next, the review focused on GDF-15 and macrophages in physiology and in several pathological contexts. However, it sometimes looks like a just a simple list of distinct studies regarding the effect GDF-15 on macrophages and/or disease outcome rather than a critical analysis of the findings that came up from these studies.

Answer: Our primary aim at this stage of the research was to make an overview for the experimental facts about GDF-15 macrophage interactions that can have impact in pathology. However, we have now added more critical evaluation in each of the sections of pathology.

As a reader, I would rather the authors highlighted and discussed thoroughly discrepancies in the different chapters, where the results of the studies are described.

Answer: Thank you for this suggestion. We now added a critical comment to each pathology section.

Additionally, I think that the authors should try to suggest potential explanations of such discrepancies (different models of liver fibrosis, different mouse strains, different approaches…) and eventually end up with a clear statement about the potential beneficial or detrimental role of GDF-15 or, in case of lack of significant results, of the need for future studies, at the end of each chapter.

Answer: At the end of each pathology chapter we added a corresponding conclusion highlighting the most important findings and need for more studies.

 In contrast, I appreciated the discussion of gaps of knowledge in the “future directions” chapter and I found tables and figures very comprehensive and clear. 

Comments on the Quality of English Language

I think that proofreading by a native English speaker might greatly improve the quality of the review and its overall impact.

Answer: We did a careful prove-read and will additionally use the editing service offered by the MDPI system

Many sentences need rephrasing to improve clarity and accuracy. I have reported a few examples below, but there are many others throughout the manuscript. 

  • Abstract “Macrophages serve as both producers and receptors of GDF-15” might be rephrased as “Macrophages are both a source and a target of GDF-15”. “Receptor” is a molecule not a cell

Answer: thank you for the suggestion. The phrase was modified, as highlighted in the abstract.

  • Lines 163-164 “In adipocytes, signal transducer and activator of transcription 6 (STAT6) triggers the expression of GDF-15 when exposed to IL-13 treatment.” Might be re-phrased as “upon IL-13 treatment, activation of signal transducer and activator of transcription 6 (STAT6) triggers the expression of GDF-15 in adipocytes” or “In IL-13-treated adipocytes, signal transducer and activator of transcription 6 (STAT6) triggers the expression of GDF-15”. Indeed, adipocytes are exposed to IL-13, while the cytokine-induced signaling cascade leads to the activation of the transcription STAT6.

Answer: We rephrased the sentence as suggested by the reviewer: “upon IL-13 treatment, activation of signal transducer and activator of transcription 6 (STAT6) triggers the expression of GDF-15 in adipocytes”

  • Lines 462. “However, the observed effects were not explained due to the binding of GDF-15 to a peripheral receptor or a macrophage receptor. “I do not understand this sentence, if the effect of GDF-15 on liver fibrosis is independent on the binding of GFD-15 to a receptor which is the potential mechanism driving GDF-15 biological activity?
  • Lines 463-464. “Additionally, Li et al. tested the systemic effect of parenteral GDF-15 reprogrammed macrophages in mice with CCl4-induced liver fibrosis”. “Systemic” is redundant because the GDF-15 is administrated by intravenous injection. Furthermore, I do not understand the meaning of the sentence. “Li et al. tested whether the adoptive transfer of GDF-15-preprogrammed macrophages in the blood circulation mitigate CCl4-induced liver fibrosis in mice” or “Li et al. tested the effect of parenterally administrated GDF-15 on macrophage reprogramming in a murine model of CCl4-induced liver fibrosis”?

Answer: Thank you for your clarity concern. The whole paragraph was change to:

“Using a GDF-15 deficient mouse model, Li et al demonstrated an increase in macrophage infiltration in liver tissue compared to non-deficient mice [131]. Moreover, the predominant macrophage infiltrating phenotype was shifted from Ly6Clow to Ly6Chi, and NF-κB signaling was hyperactivated in GDF-15 KO mice, while rGDF-15 was able to alleviate hepatic fibrosis. The authors suggested that this may be a consequence of de-creased M1 polarization and reduced expression of proinflammatory cytokines in the liver in the presence of GDF-15. However, this study did not show how GDF-15 exerts these effects in the macrophage system, whether through a peripheral receptor in a paracrine manner or through a macrophage receptor itself. Finally, Li et al. tested the effect of GDF-15 preprogrammed macrophages administered parenterally in mice with CCl4-induced liver fibrosis. This cell therapy successfully decreased the severity of hepatic fibrosis [131].”.

  • Line 474. Similarly, “ADRB1/2 is a catecholamine receptor, which is stimulated by GDF-15” means that GDF-15 binds and activates ADRB1/2 receptor or that GDF-15 upregulates the ADRB1/2 receptor?

Answer: We acknowledge this question and changed the paragraph to:

Kupffer cells are essential contributors to the progression of alcohol-induced liver damage, leading to its fibro-inflammatory stages. Kim et al. suggested that increased catecholamines play a protective role in preventing the progression of alcohol-induced hepatic fibrosis in chronic ethanol-fed mice [133]. They showed that catecholamines can in-duce apoptosis in Kupffer cells under ethanol-induced oxidative stress. This effect was shown to be mediated by beta-1 adrenergic receptor 1/2 (ADRB1/2), a catecholamine receptor whose expression in Kupffer cells was upregulated by GDF-15. They also demonstrated high levels of GDF-15 in the hepatic tis-sue in the alcohol-induced hepatic fibrosis model. The decrease on Kupffer cells prevented the accelerated fibrosis progression. These findings shed light on the mechanism by which GDF-15 functions as a stress-induced cytokine, promoting apoptosis of inflammatory Kupffer cells, thereby mitigating further hepatic damage [133].

  • Line 523. “the specific pathway surrounding this hypothesis needs to be investigated.” The molecular mechanism supporting this hypothesis needs to be investigated”

Answer: Thank you for this suggestion. The phrase was changed to: “The molecular mechanism supporting this hypothesis needs to be investigated”

  • Line 533. “The interaction of YKL-40 and GDF-15 leads to tumor invasion and suppression of the immune response of CD8+ T lymphocytes through the induction of PD-L1 in gallbladder tumor cells” might be re-phrased as “The interaction of YKL-40 and GDF-15 promotes tumor invasion and upregulates PD-L1 expression in gallbladder tumor cells, which results in the suppression of anti-tumor CD8+ T lymphocytes”.

Answer: We gratefully thank you for this suggestion. The phrase was changed to: “The interaction of YKL-40 and GDF-15 has been demonstrated to promote tumor invasion and to upregulate PD-L1 expression in tumor cells. PD-L1 binds to its receptor PD-1 on T cells, resulting in the suppression of anti-tumor activity of CD8+ T lymphocytes, thereby contributing to immune evasion [145]. Furthermore, the induced expression of PD-L1 by GDF-15 has been demonstrated in glioblastoma cells.”. 

Reviewer 2 Report

Comments and Suggestions for Authors

The manuscript provides a comprehensive review of Growth Differentiation Factor 15 (GDF-15) and its role in macrophages. The authors explore the dual roles of GDF-15 as both a cytokine produced by and acting upon macrophages. They delve into the molecular mechanisms, physiological and pathological contexts, and the therapeutic potential of GDF-15, especially in relation to inflammation, fibrosis, and cancer.

Strengths:

  1. Comprehensive Coverage: The manuscript extensively covers the literature on GDF-15, providing a detailed overview of its biological roles and mechanisms of action.
  2. Clear Structure: The review is well-organized, with sections clearly delineating the different aspects of GDF-15's interaction with macrophages.
  3. Relevance: Given the growing interest in cytokines and their therapeutic potential in treating various diseases, the topic is highly relevant.
  4. Figures and Tables: Including detailed figures and tables helps visualize complex information and summarize key points effectively.

Weakness:

1.     While the review is comprehensive, it sometimes lacks depth in discussing the precise molecular mechanisms by which GDF-15 exerts its effects.

2.     The manuscript could benefit from a discussion on the most recent advances in GDF-15 research, particularly any new therapeutic approaches or clinical trials that have emerged.

3.     There are instances of repetitive information across sections, which could be streamlined for a more concise presentation.

4.     The abstract provides a good overview but could be more concise. The key findings and significance of GDF-15 should be highlighted succinctly.

5.     The introduction sets the stage well but could briefly mention the most recent developments in the field to contextualize the review's relevance.

6.     The discussion on therapeutic potential is robust but would benefit from a more critical evaluation of the challenges and limitations faced in translating GDF-15 research into clinical practice.

Author Response

  1. While the review is comprehensive, it sometimes lacks depth in discussing the precise molecular mechanisms by which GDF-15 exerts its effects.

Answer: We primarily did not include this information about mechanism of GDF-15 action since this knowledge for macrophages is still not clarified. However, we did discuss the known molecular mechanisms exerted by GDF-15 in the pathology section. In the revised version we added brief information about the mechanism of GDF-15 action, and discussed the main limitations of the revised papers. New text fragments are added to the sections including Sepsis and inflammation, Fibrosis, Cancer and Metabolic and cardiovascular disorders.

  1. The manuscript could benefit from a discussion on the most recent advances in GDF-15 research, particularly any new therapeutic approaches or clinical trials that have emerged.

Answer: We added an update of new therapeutic approaches to the section Future directions. We also added information regarding this topic in the section Metabolic and cardiovascular disorders. To date and to our knowledge no other therapeutic approached involving macrophage/GDF-15 interaction have emerged.

  1. There are instances of repetitive information across sections, which could be streamlined for a more concise presentation.

Answer: Thank you for your suggestion. To improve this aspect, we checked for repetitive information along the manuscript and deleted, for example, content in line 336-340, 518-520 and 762-770, for clarity improvement.           

  1. The abstract provides a good overview but could be more concise. The key findings and significance of GDF-15 should be highlighted succinctly.

Answer: Additional details of GDF-15-macropahge cross-talk, as well as the new GDF-15-based therapy, have been provided in the abstract as follows:

Growth differentiation factor 15 (GDF-15) is a multifunctional cytokine that belongs to the transforming growth factor-beta (TGF-β) superfamily. GDF-15 is involved in immune tolerance and is elevated in several acute and chronic stress conditions, often correlating with disease severity and patient prognosis in cancer, metabolic and cardiovascular. Despite these clinical associations, the molecular mechanisms orchestrating its effects remain to be elucidated. The effects of GDF-15 are pleiotropic, but cell specific, and dependent on the microenvironment. While GDF-15 expression can be stimulated by inflammatory mediators, its predominant effects were reported as anti-inflammatory and pro-fibrotic. The role of GDF-15 in the macrophage system has been increasingly investigated in recent years. Macrophages produce high levels of GDF-15 during oxidative and lysosomal stress, which can lead to fibrogenesis and angiogenesis at the tissue level. At the same time, macrophages respond to GDF-15 by switching their phenotype to a tolerogenic one. Several GDF-15-based therapies are under development, including GDF-15 analogs/mimetics and GDF-15-targeting monoclonal antibodies. In this review, we summarize the major physiological and pathological contexts in which GDF-15 interacts with macrophages. We additionally discuss the major challenges and future perspectives in the therapeutic translation of GDF-15.

  1. The introduction sets the stage well but could briefly mention the most recent developments in the field to contextualize the review's relevance.

Answer: we have highlighted the recent development in the field and the need for summarizing the knowledge about the interplay between GDF-15 in macrophages in the following new paragraph in Introduction:

Several years ago, reprogramming of macrophages aroused a new wave of interest, driven primarily by the need to enhance the efficacy of conventional chemotherapy and novel immunotherapy in solid tumors [10-12]. The principles of reprogramming, initially developed for tumor-associated macrophages (TAMs), can be further utilized for therapeutic control of macrophage activity in patients with cardiometabolic diseases and in regenerative medicine. Successful reprogramming requires an understanding of the key cytokines that are able to regulate the switch from an inflammatory to a healing phenotype in macrophages and in the tissue microenvironment. The accumulated experimental data indicate that GDF-15 is such a cytokine.

  1. The discussion on therapeutic potential is robust but would benefit from a more critical evaluation of the challenges and limitations faced in translating GDF-15 research into clinical practice.

Answer: we included additional lines regarding this topic in the section Future directions